



# Trends and annual cycles in soundings of Arctic tropospheric ozone

Bo Christiansen[1], Nis Jepsen[1], Rigel Kivi[2], Georg Hansen[3], Niels Larsen[1], and Ulrik S. Korsholm[1]

[1]Danish Meteorological Institute, Copenhagen, Denmark
[2]Finnish Meteorological Institute, Helsinki, Finland
[3]Norwegian Institute for Air Research, Fram Centre, Tromsø, Norway

*Correspondence to:* Bo Christiansen (boc@dmi.dk)

**Abstract.** Ozone soundings from 9 Nordic stations have been homogenized and interpolated to standard pressure levels. The different stations have very different data coverage; the longest period with data is from the end of the 1980ies to 2014.

At each pressure level the homogenized ozone time-series have been analyzed with a model which includes both low-frequency variability in form of a polynomial, an annual cycle with harmonics, the possibility for low-frequency variability in
the annual amplitude and phasing, and either white noise or noise given by a first order autoregressive process. The fitting of the parameters is performed with a Bayesian approach not only giving the mean values but also confidence intervals.

The results show that all stations agree on a well-defined annual cycle in the free troposphere with a relatively confined maximum in the early summer. Regarding the low-frequency variability it is found that Scoresbysund, Ny Ålesund, Sodankylä, Eureka, and Ørland show similar, significant signals with a maximum near 2005 followed by a decrease. This change is
characteristic for all pressure levels in the free troposphere. A significant change in the annual cycle was found for Ny Ålesund, Scoresbysund and Sodankylä. The changes at these stations are in agreement with the interpretation that the early summer maximum is appearing earlier in the year.

The results are shown to be robust to the different settings of the model parameters such as the order of the polynomial, number of harmonics in the annual cycle, and the type of noise.

# 1  Introduction

Tropospheric ozone is a short-lived trace-gas with a life-time of 3-4 weeks in average and a following strong temporal and spatial variability. Tropospheric ozone is dangerous to human health and crops. Furthermore, tropospheric ozone is a greenhouse gas – and therefore often characterized as a short-lived climate forcer or short-lived climate component – and the increase over the 20th century has led to a considerable positive (warming) radiative forcing only exceeded by that contributed by carbon
dioxide and methane (Forster and Ramaswamy, 2007). Tropospheric ozone profiles from satellites have only been available for a decade so information about long term trends and variability mainly comes from in situ measurements such as balloon soundings.

Tropospheric ozone originates from intrusions of stratospheric air or is produced in the troposphere itself by photo-chemical processes involving precursors such as nitrogen oxides. The precursors may be of natural origin or due to anthropogenic ac-
tivities (see the review by Cooper et al., 2014). The sinks are photo-chemical processes and dry composition at the surface.



While the photo-chemical processes dominate globally, model studies (Wespes et al., 2012) indicate that in the Arctic anthropogenic pollution from the Northern Hemisphere is the dominant source of ozone from the surface to 400 hPa and that the stratospheric influence is the main contribution at pressures less 400 hPa. The anthropogenic sources may either be formed in situ or transported to the site of reaction. In particular, during summer emissions from fires in Russia and North America

impact the tropospheric ozone in the Arctic. Nitrogen oxides are considered especially important in this respect and apart from originating from anthropogenic activities they may also be formed in lightning processes (Cairo et al., 2010). The influx from the stratosphere may be caused by tropopause foldings as has been demonstrated using backwards trajectory calculations (Sørensen and Nielsen, 2001). Synoptic scale processes as represented by the 250 hPa geopotential height have also been successfully linked to the recent ozone increases in the lowermost stratosphere (Harris et al., 2008). Analysis of observations in the

2008 International Polar Year (Ancellet et al., 2016) indicates that stratosphere-troposphere exchange is larger over Greenland than over Canada.

     In the 20th century there has globally been a general increase in tropospheric ozone in qualitative agreement with the increasing levels of nitrogen oxides from pollution. In the last part of the 20th century ozone level stabilized over Europe and North America (Guicherit and Roemer, 2000). See also the reviews of Cooper et al. (2014) and Hartmann et al. (2013). A

flattening of the trend is also seen in other regions over the last 10–15 year – although with many regional differences – and it is likely that this is at least partly due to the fact that the emission of precursors has been curbed (Oltmans et al., 2013). It should be noted that changes in tropospheric circulation patterns also may play a role (Lin et al., 2014).

     In the northern hemisphere (NH) troposphere ozone peaks in the late spring or summer (e.g., Parrish et al., 2013; Cooper et al., 2014). The spring-summer peak is often attributed to enhanced photo-chemical production (Monks, 2000) and the latest occur-

ring of the peak is often found in the most polluted continental regions. However, it has also been argued that the stratosphere-troposphere exchange may play a role.

     There has been found evidence for that the seasonal cycle of tropospheric ozone in the NH mid-latitudes has changed so that the peak now appear earlier than twenty years ago (Parrish et al., 2013). Parrish et al. (2013) finds in a study of 5 stations that the change in the peak occurrence is 3-6 days per decade since 1970. Cooper et al. (2014) extended the analysis including

additional sites and confirmed that there is a general shift although not observed at all sites. Possible reasons for the changes in the seasonal cycle are changes in atmospheric patterns and emissions. Cooper et al. (2014) also called for additional analysis including, e.g., the polar regions.

     In the Arctic balloon soundings are relatively scarce and the measurement periods vary from station to station. The longest data series are from Resolute, Canada (Tarasick et al., 2005). In the European sector of the Arctic and over Greenland the

ozonesondes have been flown since late 1980s (Kivi et al., 2007). Accordingly, the reported long-term changes in tropospheric ozone are scattered. Logan et al. (1999) found decreasing tropospheric ozone at Resolute, Canada in the period 1970 – 1996. Also Fioletov et al. (1997) and Randel and Wu (1999) have reported ozone decreases at Resolute. Negative trends in tropospheric ozone over Canada in the period 1980–1993 were also reported by Tarasick et al. (1995) and Oltmans et al. (1998). Later, Tarasick et al. (2005) also noted that when the period 1991–2001 is considered the trends are positive. Oltmans et al.





(2013) found for 3 stations in the arctic Canada that negative trends in the beginning of the period 1980-2010 had been neutralized by positive trends later in the period.

Kivi et al. (2007) studied the variations in ozone profiles using ozonesonde observations from seven northern high-latitude stations from 1989 to 2003. In the free troposphere they found a statistically significant increase of 11 % in this period with largest values in January to April, the period of greatest interannual variability. They attributed the observed change to the combined increase in the stratosphere-troposphere exchange and the transport of precursors towards the higher latitudes.

Here, we investigate ozone variability over 9 northern high-latitude stations, with an emphasis on the measurements made over Northern Europe and Greenland. We focus on the low-frequency variability and on the changes in the annual cycle for which previous results in the Arctic are scarce. The present study includes recent ozonesonde measurements obtained in the period from the early 2000s to 2014, which have not been analysed in details before. This results in a 27-year dataset for the longest record. We include ozonesonde data from Bear Island, Ørland, and Gardermoen that have not been considered in the previous studies of tropospheric ozone. The measurements are homogenized according to current recommendations. The ozone time-series from the individual stations are analyzed with a model, which includes both low-frequency variability and the annual cycle with higher harmonics. The potential for low-frequency variability is implemented both as a general polynomial trend and time-varying annual amplitudes and phases. The noise is either white or given by a first order autoregressive process. The model is nonlinear and may include a large number of parameter. The fitting of these parameters is performed with a Bayesian approach. The Bayesian approach gives us mean values and uncertainties not only of the parameters but also on derived quantities such as temporal differences and annual cycles. This approach naturally handles strongly irregular sampled time-series including extended periods without data and is therefore favorable for the analysis of ozone time-series.

## 2 The data and method

### 2.1 Ozonesonde data

The ozonesonde is an electrochemical device containing two electrode chambers: an anode chamber filled with potassium iodide saturated phosphate buffer and a cathode chamber filled with same phosphate buffer containing a well-defined concentration of potassium iodide (Kivi et al., 2007; Smit, 2014). During ascent through the atmosphere a constant volume pump is drawing atmospheric air through the cathode chamber. The content of ozone in an air sample is reacting with the potassium iodide and gives rise to a current proportional to the ozone amount. The electrode chambers and the pump is installed in a Styrofoam box for insolation purposes. To keep the buffer liquids from freezing during ascent a simple heater element is keeping the temperature in box at 10-25°C. A thermistor is sensing the actual temperature inside the box. On the outside of the Styrofoam box a regular radiosonde is mounted. The radiosonde is measuring pressure, temperature, humidity, wind speed, and wind direction during ascent. The ozone current and the box temperature is via an interface transmitted to a ground receiver along with the radiosonde parameters. The ozonesonde and the radiosonde are lifted with a helium or hydrogen filled meteorological balloon. At best the balloon may reach an altitude at 35-40 km. The typical vertical resolution is around 10 m using 2 seconds intervals for sampling. However, the effective vertical resolution is of the order of 100-150 m, given that the response time of





the ozone sensor is 20-30 s. Uncertainty of the ozone measurements by electrochemical sondes in the stratosphere is about 5 % (Deshler et al., 2008; Hassler et al., 2014).

Different types of ozonesondes have been in use over the years, the primary two types being manufactured by EnSci and Science Pump. Both types are constructed as described above. For each ozonesonde type there is a recommended composition of the anode and cathode solutions in use. Problems arise with a change to a different brand of ozonesonde. Historically many launches have been made using sensing solution recommended for Science Pump ozonesondes in case of switching to the use of EnSci type ozonesondes. To investigate the difference between the two sonde types and sensing solutions a number of in situ measurements have been performed in the laboratory (Smit et al., 2007) and in the field (Kivi et al., 2007; Deshler et al., 2008). These measurements have resulted in the current recommendations for the ozonesonde preparations (Smit, 2014). In this work ozonesonde data were homogenized according to the recommended transfer functions for data homogenisation (Deshler et al., 2017).

The geographic distribution and the covered time period for the included stations are summarized in Table 1. The number of soundings for each station as a function of year is shown in Fig. 1. The longest times-series span the period from the late 1980ies to 2014. The time-series of Bear Island, Gardermoen, and Ørland are particular brief spanning less than 10 years. In general the soundings are highly irregular timed with occasional years with very few or none soundings. We also note that the details vary a lot among the stations. The average yearly number of soundings are largest (around 90) for Ny Ålesund and lowest for Thule (around 20). There are in general more soundings in winter and spring than in summer and autumn (Table 1 shows the seasonal average of number of soundings disregarding years without soundings). This is due to the frequent ozonesonde campaigns to investigate the stratospheric vortex ozone depletion during the winter/spring season (Rex, 1993; von der Gathen et al., 1995; Manney et al., 2011).

For each station and for each homogenized ozone sounding the ozone has been interpolated to standard pressure levels between 900 and 10 hPa (900, 800, ... 300, 250, ... 100, 80, 70 ... 10 hPa.). The resulting ozone fields are shown as function of time and pressure in Fig. 2 for each station. As expected there is a maximum on the lower stratosphere. Here and in the rest of the paper ozone partial pressure is measured in millipascal (mPa). Time series of the free tropospheric ozone at 500 hPa are shown in Fig. 3 (black dots). We already here note that these ozone records show a background level of 2-4 mPa and that the ozone records have large annual cycles and a considerable amount of scatter.

## 2.2 Model description

At each pressure level we want to model the temporal development of ozone. We are particularly interested in potential low-frequency trends, the annual cycle, and changes in the annual cycle. We therefore use a model that contains a trend, an annual cycle and noise. The model has the form

$$y = \lambda_0 + \lambda_1 t + \lambda_2 t^2 + ...a_1 \sin(2\pi t + \theta_1) + a_2 \sin(2\pi 2t + \theta_2)... + \xi.$$

where $y$ is the ozone and $t$ is the time (in years). Note that the amplitudes, $a_i$, and phases, $\theta_i$, may depend on time as detailed below.





The model has the following properties:

- The trend consists of a constant $\lambda_0$, a linear trend $\lambda_1 t$, and higher order polynomials up to $\lambda_{n_{pol}-1} t^{n_{pol}-1}$.

- The annual cycle consist of a sum of $n_{cyc}$ sinusoidals, $a_i \sin(2\pi i t + \theta_i)$, with frequencies $1, 2, 3 \ldots n_{cyc}$. The higher harmonics allow the seasonal cycle to be asymmetric. The amplitudes and phases of the cycles have trends with $n_{tr}^a$ and $n_{tr}^\theta$ terms: $a_i = a_{i,0} + a_{i,1}t + \ldots a_{i,n_{tr}^a} t^{n_{tr}^a}$, $\theta_i = \theta_{i,0} + \theta_{i,1}t + \ldots \theta_{i,n_{tr}^\theta} t^{n_{tr}^\theta}$. This allows the annual cycle to change over time.

- The noise is either independent Gaussian with variance $\sigma^2$ or an first order auto-regressive process (AR1) with coefficient $\theta$ and variance $\sigma^2$.

- Then, the model totally includes $2 + n_{pol} + n_{cyc}(1 + n_{tr}^a + n_{tr}^\theta)$ parameters under AR1 noise and one less under Gaussian noise.

The model is nonlinear and includes a considerable number of parameters. The data (Fig. 1) are irregular samples with strong changes in the number of soundings over time but also with a strong seasonal cycle in the number of soundings. Calculating monthly or annual means followed by an estimation of the annual cycle and trends from these means – as done in some previous studies – is sub-optimal. It will, in particular, make the uncertainty difficult to estimate trustfully.

We therefore choose a Bayesian approach for interference (see, e.g., Gelman et al., 2004). The Bayesian approach does not require regular temporally gridded data but can work directly with the original sampling. Bayesian approaches are becoming more frequent in many different area of atmospheric and climate sciences (see, e.g., Hasselmann, 1998; Berliner et al., 2000; Haslett et al., 2006; Huang et al., 2011; Tingley and Li, 2012; Aldrin et al., 2012; Christiansen, 2014; Olson et al., 2016). Samples from the posterior are obtained by a simple Metropolis-Hastings algorithm (Brooks et al., 2011) and we assume flat priors for all parameters. This approach not only produces ensembles of all parameters but also of all derived quantities such as trends, annual cycles, and changes in the annual cycles. These ensembles give the posterior distributions of the quantities under consideration and from these distributions we calculate and report the posterior mean and the 95 % confidence intervals (or credible intervals as they are called in the Bayesian literature). Thus, this approach can provide mean and confidence intervals for, i.e., the difference of the annual cycle between two periods. We produce a large ensemble (20000 members) of the posteriors and make sure that the process has converged. We discard the first half of the ensemble to avoid transients.

## 3 Results

Given the large differences in data coverage among the different stations we can not expect that all station can provide sufficient information to constrain models with a high number of parameters. We therefore begin the analysis with a simple version of the model including only the polynomial trend and a fixed annual cycle. In subsection 3.1 this model is used to study the long term mean and the trends, and in subsection 3.2 it is used to study the mean annual cycle. In subsection 3.3 we extent the model to include trends in the amplitudes and phases of the annual cycle so that changes in the annual cycle can be studied. We





only apply the extended model to the four stations with the best data coverage. In all subsections we begin by considering the 500 hPa level before we proceed to other levels of the troposphere. As mentioned the Bayesian approach not only gives point values but the whole posterior distributions so we are able to produce confidence intervals for all the studied quantities.

## 3.1 Mean and trends

Figure 3 shows for each station at 500 hPa the raw data (black points), the posterior mean of the non-stochastic part of the model, i.e., the polynomial part and the annual cycle (cyan), and the posterior mean of the polynomial part of the model (green) alone. The model includes a third order polynomial ($n_{pol} = 4$) and two components in the annual cycle ($n_{cyc} = 2$). The model does not include trends in the amplitudes and phases of the annual cycle and the noise is assumed white.

It is obvious that the Bayesian procedure has produced reasonable fits dominated by an annual cycle and including a weak
inter-decadal variability. It is also obvious that there is a considerable residual scatter at all stations. This scatter is the expression of dynamical and chemical processes in the atmosphere as well as measurement noise.

Figure 4 shows both the mean polynomial part of the model (the cyan curve in Fig. 3) and its 95 % confidence interval for each point in time at 500 hPa. For all stations the long term background value is around 3 mPa and the polynomial part is relatively flat with some weak low-frequency variability. The 95 % confidence intervals are quite large relative to the low-
frequency variability. This mainly reflects the data coverage but the confidence intervals also increases near the beginning and end of the time-series where data are limited because of the asymmetry. For Scoresbysund, Sodankylä, Ny Ålesund, and Eureka some significant albeit weak low-frequency variability can be discerned. At Scoresbysund the ozone partial pressure increases until a maximum is reached near 2007 followed by a weak decrease. Ny Ålesund shows similar behavior but now with the maximum around 2003. Sodankylä also shows a decrease in recent years with a maximum around 2005. However,
Eureka shows a qualitative different variability with a strong increase from 1993 to 2000 followed by a quiet period until 2008 after which it again increases. At Thule, Bear Island, Gardermoen and Lerwick no significant trends are found.

While the discussions above dealt with the 500 hPa layer we now consider all layers in the troposphere. Figure 5 shows the long term mean as function of height. We see that the form of the vertical variations are identical for all stations. At the lowest level, 900 hPa, the mean ozone level is between 3 and 4 mPa for all stations. The ozone content then decreases with height
throughout the troposphere until a well-defined minimum of approximately 2.5 mPa is reached around 300 - 400 hPa. The ozone content then increases fast with height when the stratosphere is reached. Note that in the troposphere it is discernible that stations at lowest latitude have larger ozone mixing ratios.

The contour plots in Fig. 6 show the anomalies at each level, i.e., the deviations from the long term mean (the right hand plots in each panel show the long term mean as in Fig. 5). Shaded areas indicates regions where the anomalies are significantly
different from zero, i.e, where the ozone content can be considered different from the long term mean. In agreement with the results at 500 hPa we do not find much significant long term variability at Thule, Bear Island, Gardermoen and Lerwick. In particular for Bear Island and Gardermoen this might be connected to the brief span of the observations. At the other stations – Scoresbysund, Sodankylä, Ny Ålesund, Eureka, and Ørland – we find a consistent and significant signal throughout the troposphere. This signal is in general equivalent barotropic in the troposphere with the same sign at all heights and values



that decreases with height. There is a general agreement at these stations that a significant maximum was reached in the years around 2005 although the exact year of the maximum varies. This is in general agreement with the discussion above about the variability at 500 hPa. The signal is weak or absent at the tropopause level but note also that a strong signal of the same sign as in the troposphere is found in the lower stratosphere.

## 3.2 Mean annual cycle

For each station Fig. 7 shows both the mean annual cycle as well as the 95 % confidence interval for each day of year at 500 hPa. The annual cycle has a strong similarity for all stations. It has a minimum in winter, a maximum in early summer and a peak-to-peak amplitude of approximately 1 mPa. We also note that the annual cycle would not be well modelled with a single sinusoidal as the early summer peak is more temporal confined than the winter minimum. The widths of the 95 % confidence intervals reflect the data coverage and are largest for Thule, Gardermoen, Ørland, and Bear Island.

The mean annual cycle as function of height is shown in Fig. 8 for each station. The annual cycle is rather similar for all stations consistent with the results for 500 hPa. For most stations there is a clear change of the phase of the annual cycle with height; the spring/summer maximum appears earlier at the lower levels than in the middle of the troposphere. This phase change is typical a couple of months. In the lower stratosphere the annual cycle again has the maximum earlier in the year. The amplitude of the annual cycle is relatively constant with height.

At the near-surface at 900 hPa there is some evidence for a qualitatively different annual cycle with a secondary maximum in autumn. This is observed for Ny Ålesund, Thule, and Eureka.

### 3.3 Changes in the annual cycle

We saw in the last section that the annual cycle was well modelled and almost identical for all stations. This provides some hope for that we have enough information to detect potential changes in the annual cycle. We limit the following analysis to the four stations with best data coverage: Scoresbysund, Sodankylä, Ny Ålesund, and Eureka. We now extent the model from the last section by setting $n_{tr}^a = n_{tr}^\theta = 2$ and thereby allowing both the amplitudes and the phases of the annual cycle to vary in time like a second order polynomial.

The results at 500 hPa are shown in Fig. 9, where the annual cycles averaged over 1995-2000 and 2007-2012 are shown together with their 95 % confidence intervals for each day of the year. It should be noted that there are large uncertainties connected to the changes in the annual cycles. The only significant change is found at Ny Ålesund which shows a slight, significant decrease from 0.9 to 0.8 in the peak-to-peak amplitude. There also seems to be a slight change in the phase with the maximum appearing a little (20 days) earlier in the later period. For the other stations there is very little and insignificant change in the amplitude and phase of the annual cycle at 500 hPa.

The differences between the mean annual cycles over 2007-2012 and 1995-2000 are shown as function of pressure level in Fig. 10. The significant change found at Ny Ålesund at 500 hPa seems consistent with other levels in the troposphere for this station. Some significant changes are now also found for Scoresbysund and Sodankylä. These changes consist of an amplification of the increasing spring branch of the annual cycle and weakening of the summer maximum. Thus, the changes





in the annual cycles at Ny Ålesund, Scoresbysund and Sodankylä have the same sign and patterns. Together this is consistent with the notion of the summer maximum appearing earlier in the year.

## 4 Robustness of the results

Our model allows for many different settings of the parameters and it is not obvious which setting that is the optimal choice.
We have, for example, in the previous discussion restricted ourselves to model setups with white noise.

In this section we briefly discuss the robustness of the results to changes in the parameters of the model. We will restrict the presentation to Scoresbysund for the low-frequency variability and to Ny Ålesund for the changes in annual cycle, but similar results are found at other stations.

The upper panels in Fig. 11 show the polynomial part of the model for Scoresbysund as function of height for model settings with either white noise or AR1 noise. The model settings also include trends in the annual cycle which was not the case in Fig. 6 We observe that all three model settings agree on the shape of the low-frequency variability and, in particular, that they agree on the maximum obtained around year 2005.

The lower panels in Fig. 11 show the difference in mean annual cycles over 2007-2012 and 1995-2000 for Ny Ålesund for two different settings which include a different number of seasonal harmonics (also compare bottom right panel in Fig. 10). Again we observe that all model settings agree on the strength and pattern of the change in the annual cycle.

These results are typical for the stations with best data coverage. Some sensitivity is seen for stations with large gaps between soundings. It should also be noted that at the levels from 300 hPa and above the residuals are strong and are positively skewed. This behavior is probably due to the proximity to the stratosphere and the positive excursions related either to variation of the tropopause height or to intrusions of ozone rich stratospheric air into the troposphere.

## 5 Conclusions

We have analyzed ozone long term sounding records from 9 Nordic stations. The different stations have very different data coverage. The longest period with data is from the end of the 1980ies to 2014. The ozonesonde data were homogenized according to the recent, recommended transfer functions. We interpolated the homogenized series to standard pressure levels and in the following analysis we focused on the tropospheric levels. We applied a model which includes both a low-frequency variability in form of a polynomial, an annual cycle with harmonics, the possibility for low-frequency variability in seasonal amplitude and phasing, and noise which could be either white or a first order autoregressive process. The fitting of the parameters were performed with a Bayesian approach not only giving the posterior mean values but also 95 % confidence intervals. This approach is appropriate for strongly scattered data such as the ozone soundings. It can deal with data-gaps and makes use of all the information in the data in contrast to methods based on producing monthly averages.

Our main findings are:



- The long term averages have the same profile for all stations. The mixing ratios decrease with height from the largest values of 3-4 mPa at the lowest layer to a well-defined minimum around 400 hPa.

- All stations agree on a well-defined annual cycle in the free troposphere with a relatively confined maximum in the early summer. While the amplitude of the annual cycle does not vary much with height in the troposphere the spring/summer maximum appears somewhat (about 50 days) earlier in the lowest layers compared to the middle troposphere.

- Regarding the low-frequency variability we find that Scoresbysund, Ny Ålesund, Sodankylä, Eureka, and Ørland show a consistent and significant structure with a maximum near 2005 followed by a decrease. This signal is equivalent barotropic with an amplitude that decreases with height.

- Some changes in the annual cycle were found for Ny Ålesund, Scoresbysund and Sodankylä with the most significant changes found for Ny Ålesund. The changes are consistent between the three stations – although there are differences in the vertical profile of the changes – and are in agreement with the notion of the summer maximum appearing earlier in the year.

- The results were shown to be robust to the different settings of the model parameters such as the order of the polynomial, number of harmonics in the annual cycle, and type of noise.

The significant maximum at Scoresbysund, Ny Ålesund, Sodankylä, Eureka, and Ørland around 2005 and the following decrease have not been reported before regarding observations in the free troposphere and the Arctic. Previous work (Kivi et al., 2007) covering data from 1989-2003 suggests a linear increase in the free troposphere of about 11 % consistent with our observations for Thule, Scoresbysund, Ny Ålesund, Eureka, Sodankylä, and Ørland. Scoresbysund, Eureka, Ny Ålesund, and Sodankylä were also included in the study by (Kivi et al., 2007). The observed change was suggested to be due to changes in the Arctic Oscillation. Also Tarasick et al. (2005) found positive trends for Canadian stations in the period 1991–2001 in contrast to the negative trends found when the longer period 1980–2001 is considered. Oltmans et al. (2013) did not find any overall trends in tropospheric ozone for 3 stations in the Canadian Arctic in the period 1980–2010; declines in the beginning of the period have rebounded. Here, we did not see any negative trends before year 2001 except perhaps for the brief series at Bear Island.

Our finding that ozone peaks in spring/summer is in agreement which what is found for the NH (Parrish et al., 2013; Cooper et al., 2014). The change in the annual cycle so that the peak now appears earlier in year has not been reported before for the Arctic but is in agreement with what is found for mid-latitudes (Parrish et al., 2013; Cooper et al., 2014), although significant changes are not found for all stations.

The decrease in Arctic tropospheric ozone since 2005 may be explained by the corresponding decrease in nitrogen oxide level observed in the mid-latitude Europe where current levels now are down to 50 % of 1990 level (European Environment Agency, 2014). Nitrogen oxide is an important precursor for the production of tropospheric ozone, but this will still require transport of this species from Europe to the Arctic. Therefore the change in free tropospheric ozone in the Arctic may reflect changes in both precursors and in transport, while possible changes in the stratosphere-troposphere exchange should be also considered.



*Data availability.* The ozone soundings can be downloaded from the World Ozone and UV database at Toronto (http://www.woudc.org) and from the NDACC database (http://www.ndsc.ncep.noaa.gov/data/)

*Competing interests.* No competing interests are present.

*Acknowledgements.* We thank David Tarasick (Eureka), Peter von der Gathen (Ny Ålesund), and Dave Moore (Lerwick) for the ozone sounding data. This study was supported by the NMR KOL group (project no. NMR KOL 1402). Research at FMI was also supported by an EU Project GAIA-CLIM, the ESA's Climate Change Initiative programme and the Ozone_cci subproject in particular.



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





**Table 1.** The stations included in the study

| Station name and country | Latitude | Longitude | Period | # soundings, whole year winter/summer/spring/autumn |
|---|---|---|---|---|
| Eureka (CA) | 80.1° N | 86.4° W | Nov 1992 - Sep 2011 | 65 25/12/19/11 |
| Ny Ålesund (DE) | 78.9° N | 11.9° E | Jan 1992 - Sep 2014 | 88 34/15/26/15 |
| Thule (Pituffik) (DK) | 76.5° N | 68.7° W | Oct 1991 - Nov 2013 | 21 12/3/8/4 |
| Bear Island (N) | 74.3° N | 19.0° E | Oct 1988 - Apr 1997 | 39 15/9/14/8 |
| Scoresbysund (DK) | 70.5° N | 22.0° W | Jan 1989 - Dec 2013 | 55 17/13/14/13 |
| Sodankylä (FI) | 67.4° N | 26.7° E | Mar 1988 - Dec 2013 | 67 24/13/18/13 |
| Ørland (N) | 63.7° N | 9.6° E | Nov 1994 - Mar 2007 | 25 10/5/8/5 |
| Gardermoen (N) | 60.2° N | 11.1° E | Oct 1990 - Feb 1998 | 35 16/6/15/6 |
| Lerwick (UK) | 60.1° N | 1.2° W | Feb 1992 - Dec 2013 | 49 19/10/14/12 |

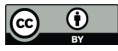



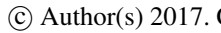

**Figure 1.** Timing of soundings. Each dot represents a sounding reaching at least 250 hPa.




**Figure 2.** Ozone partial pressure [mPa] as function of time and pressure for the 9 stations.

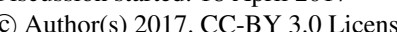



**Figure 3.** Ozone at 500 hPa (partial pressure mPa) for the 9 stations. Observations (black), model mean fit (cyan) and polynomial part of the model (green) as function of time at 500 hPa. Note the different time-ranges used for the different stations. Model settings: $n_{pol} = 4$, $n_{cyc} = 2$, $n_{tr}^{a} = n_{tr}^{\theta} = 0$, and white noise.




**Figure 4.** The polynomial part of the model as function of time at 500 hPa. Green curve shows posterior mean, black curves indicate the 95 % confidence intervals for each point in time. Note the different time-ranges used for the different stations. Model settings: $n_{pol} = 4$, $n_{cyc} = 2$, $n_{tr}^a = n_{tr}^\theta = 0$ and white noise.





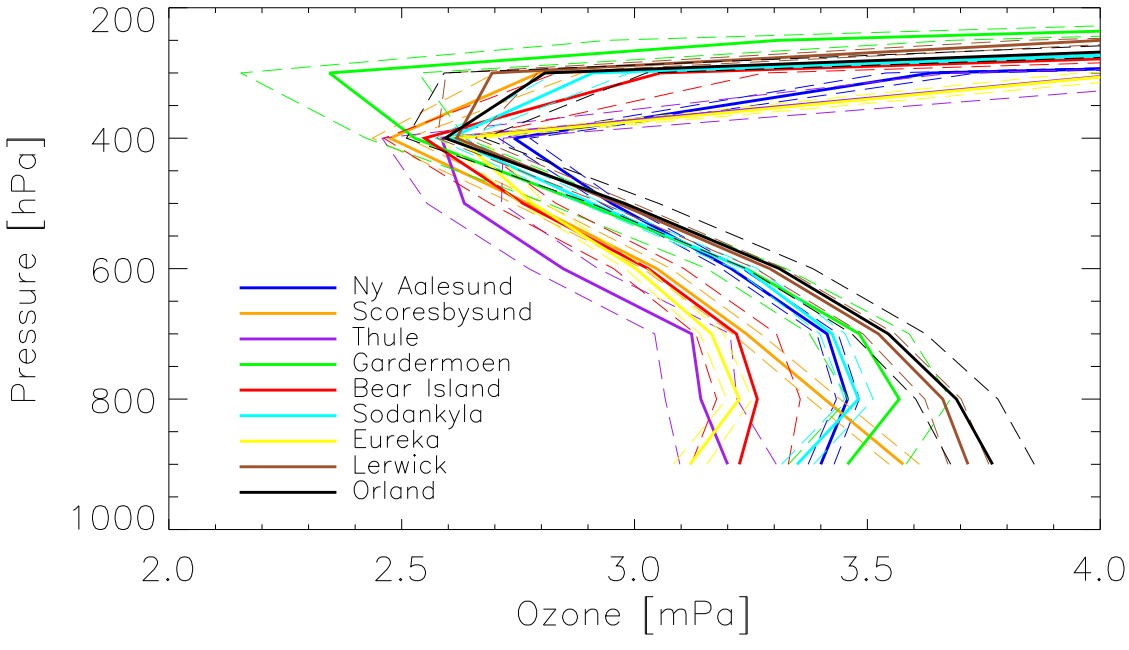

**Figure 5.** The long term mean as function of pressure (solid curves). Dashed curves indicate the 95 % confidence intervals. Model settings: $n_{pol} = 4$, $n_{cyc} = 2$, $n_{tr}^a = n_{tr}^\theta = 0$, and white noise.



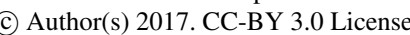



**Figure 6.** The polynomial part of the model as function of time and pressure. The temporal means are shown in the panel to the right as function of height. The contours show the anomalies with respect to this mean. Shaded regions are where the anomalies are statistically different from the temporal means at 1 and 5 % levels. Model settings: $n_{pol} = 4$, $n_{cyc} = 2$, $n_{tr}^1 = n_{tr}^\theta = 0$ and white noise.



**Figure 7.** The annual cycle as function of day of year at 500 hPa. Black curve shows posterior mean, colored curves indicate the 95 % confidence intervals for each day of year. Model settings: $n_{pol} = 4$, $n_{cyc} = 2$, $n^1_{tr} = n^\theta_{tr} = 0$ and white noise.




**Figure 8.** Mean annual cycle as function of pressure level. Model settings: $n_{pol} = 4$, $n_{cyc} = 2$, $n_{tr}^{a} = n_{tr}^{\theta} = 0$, and white noise.





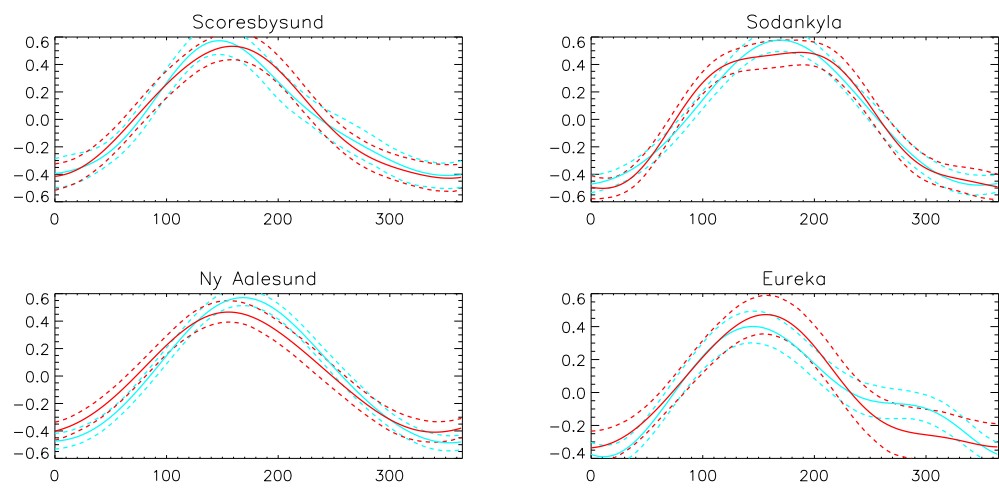

**Figure 9.** Average annual cycles over 1995-2000 (cyan) and 2007-2012 (red) at 500 hPa. Full curve is the posterior mean, dashed curves indicate the 95 % confidence intervals. Model settings: $n_{pol} = 4$, $n_{cyc} = 3$, $n_{tr}^a = n_{tr}^\theta = 2$, and white noise.



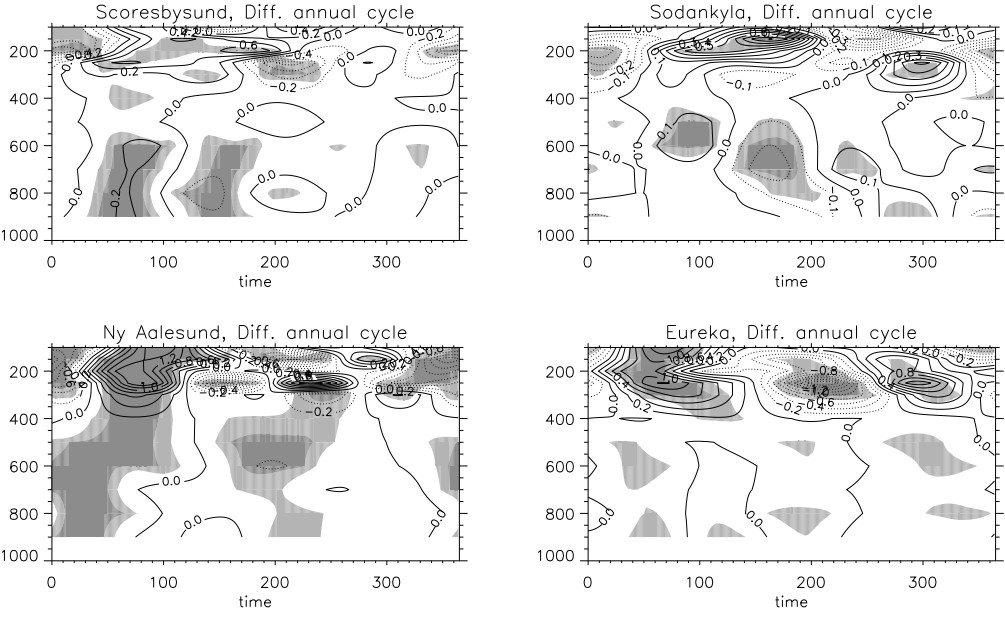

**Figure 10.** Difference between average annual cycles over 2007-2012 and 1995-2000 (i.e., average over 1995-2000 subtracted from average over 2007-2012) as function of pressure level. Shaded regions are where the anomalies are statistically different from the temporal means at 1 and 5 % levels. Model settings: $n_{pol} = 4$, $n_{cyc} = 3$, $n_{tr}^a = n_{tr}^\theta = 2$, and white noise.





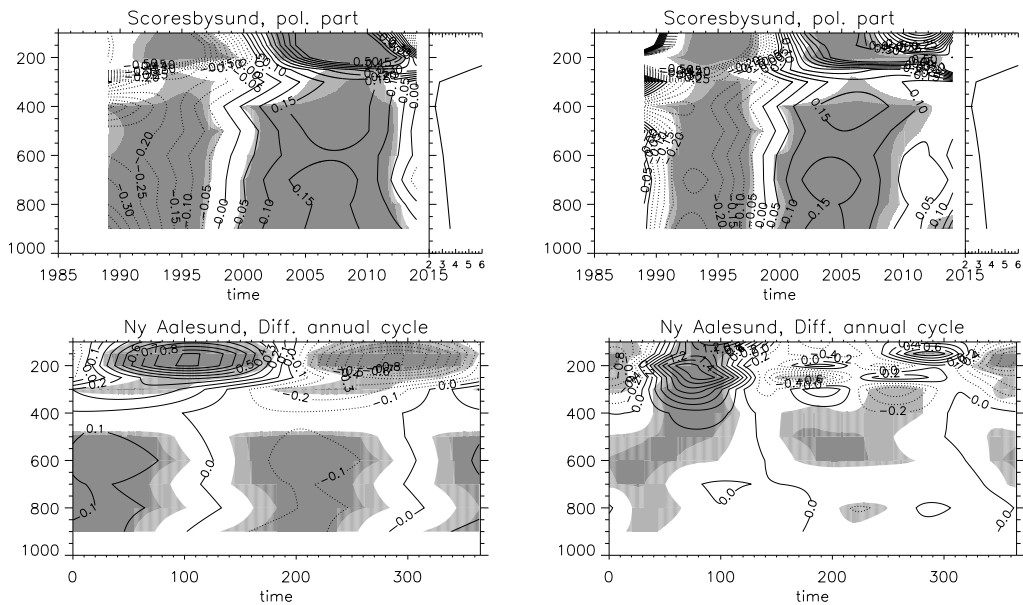

**Figure 11.** Top: The polynomial part of the model as function of time and pressure for Scoresbysund. The models are (left): $n_{pol} = 4$, $n_{cyc} = 3$, $n_{tr}^a = n_{tr}^\theta = 2$, and white noise, (right) $n_{pol} = 5$, $n_{cyc} = 3$, $n_{tr}^a = n_{tr}^\theta = 3$, and AR1 noise. Compare also to top right plot in Fig. 6 which does not include trends in annual cycle ($n_{pol} = 4$, $n_{cyc} = 2$, $n_{tr}^1 = n_{tr}^\theta = 0$ and white noise). Bottom: Difference between average annual cycles over 1995-2000 and 2007-2012 as function of pressure level for Ny Åle sund. Left: $n_{pol} = 4$, $n_{cyc} = 1$, $n_{tr}^a = n_{tr}^\theta = 1$, and AR1 noise. Right: $n_{pol} = 5$, $n_{cyc} = 3$, $n_{tr}^a = n_{tr}^\theta = 3$, and AR1 noise. Compare also to lower left panel in Fig. 10.