# Peer review of "Trends and annual cycles in soundings of Arctic tropospheric ozone"

_Atmospheric Chemistry and Physics, 2017_

## Referee Comment (RC1) · Anonymous Referee #1 · 19 May 2017

This paper analyzes a homogenized ozonesonde data set from nine stations in the Arctic region. The homogenization procedure is the one recently proposed by the WMO/GAW ozonesonde activity group. The focuses of the data analysis are the annual cycles and long-term low frequency variability in the past ∼25 years, by using a polynomial model with these components. There were similar studies (but published several years ago) for northern-hemisphere mid and high latitudes, but this is the first study looking at the Arctic region, with the homogenized data set and with the period up to more recent years. I think that the paper would be suitable for publication in Atmospheric Chemistry and Physics, with some more explanation on the homogenization procedure and on the model as explained below.

1. For the homogenization, please clarify whether data from all 9 stations were homogenized by the authors or some were homogenized by the researchers listed in

the Acknowledgements section. Although the guidelines of the homogenization procedure are documented by Deshler et al. (2017), there should be several details at each station. Will the authors prepare a separate document on the details of the data homogenization at each station? Also, at the Data availability section, the authors write that the data can be obtained from WOUDC and NDACC website. Are both the original and homogenized data available from there? For this paper, I think that an additional figure showing the change points of the ozonesonde type and solution type at all the nine stations would be very useful, in particular when the long-term trends for the same station obtained by previous studies and by this study may differ.

2. I am not sure to what degree the model description should be detailed, but I think there are few more things that the authors can do to convince the readers of the goodness of the fit. (For your information, I only know multiple regression analysis (e.g., Chapter 8.4 of von Storch and Zwiers, 1999: Statistical Analysis in Climatic Research, Cambridge Univ. Press, Cambridge, UK, 484 pp.) where coefficients are obtained with the least squares method and the statistical significance test is made based on the residual time series $x_i$.) At least, please add some more explanation on (1) how (or which part of the model) to use the two different noise models and on (2) the Bayesian approach by contrasting it with other approaches. One potential way might be to provide (the essential part) of the code as a supplement (or, specify the paper that is most relevant in terms of authors' actual calculations for this paper.) More important is to simply show (for one or two cases) that the residual time series has no significant trends/low-frequency variability, i.e., show no strange behavior (perhaps just in the authors' response or in another supplement), so that the readers can see that the model is reasonable.

---

## Referee Comment (RC2) · Anonymous Referee #2 · 23 May 2017

This paper presents an analysis of the annual cycle and trends of Arctic tropospheric ozone from ozonesonde measurements at 9 stations with relatively long records. Measurements from different types of ozonesondes have been adjusted to produce homogenized time series. The time series are analyzed using a Bayesian statistical model to evaluate the annual cycle, trends and changes in the annual cycle. Overall consistent results are found among the different stations, highlighting a clear annual cycle (with an interesting vertical structure), plus small trends and changes in the annual cycle. Overall this is a careful and straightforward analysis that contributes new knowledge on the behavior of Arctic ozone (although the small derived changes must be interpreted in light of geographic location and relatively short time series). The paper is well written and is appropriate for publishing in ACP. I have several comments and suggestions that the authors might consider in revision.

1) It might be good to include a map showing the station locations. All of the stations analyzed here are from a relatively narrow longitudinal range (∼90 W to 30 E), and so it might not be too surprising to see similar seasonal and long-term variability among stations (especially for the closely-spaced stations in northern Europe). I appreciate that long-term data records do not exist for other longitudes. However, there are two Canadian Arctic stations with long records not included here (Alert and Resolute), for which data is easily available. Is there any reason not to include these in this analysis just for completeness? (and focus the seasonal cycle and trend discussions a little more on regional differences, as noted below).

2) For readers not familiar with Bayesian analyses, could you describe the procedures in a little more simple language? For example, 'Samples from the posterior are obtained by a simple Metropolis-Hastings algorithm and we assume flat priors for all parameters' (p. 5) might not be clear to everyone.

3) The isolation of the annual cycle at all of the stations is a nice result, but can the authors try to explain what is controlling this interesting vertical structure? This appears to me to be some combination of in situ generation / transport at lowest levels, along with downward transport from the stratosphere at upper levels (with influences down to 500 or 600 hPa). It might be useful to include a seasonally-varying tropopause in Fig. 8. Also, note that early winter low-level ozone maximum at Thule and Eureka (Fig. 8) may represent a regional behavior, different from the European sector (all other stations). Including Alert and Resolute could help clarify this behavior.

4) Figures 4 and 6 would be improved if all of the stations used the same time axis (for easier comparisons), and maybe only include the stations with long time series in Fig. 6. It looks like there might be some systematic differences between the polynomial fit in the European region (larger maxima ∼2005) versus those at Thule and Eureka (and useful to include Alert and Resolute if possible). What does the coherence between the troposphere and stratosphere (or lack thereof) in Fig. 6 imply for causes of the long-term variations?

[Figure]

5) The interesting seasonal cycle changes seen in Fig. 10 are relatively small and have different altitude behavior among the three European sector stations (and are not evident at Eureka). From Fig. 9 it looks like the 95% uncertainty levels overlap for almost the entire year at Ny Alesund; how is this consistent with the 1% significant differences at 500 hPa shown in Fig. 10? I think adding results from Alert and Resolute (comment 1 above) might help identify if the small seasonal cycle changes are a north-European regional effect or larger-scale result.

6) Statement on p. 8, l. 15: I disagree that the patterns in Fig. 11 agree on the 'strength and pattern of the change in the annual cycle'. The vertical structure is very different between the white noise and AR1 results in Fig. 11. This is useful information, but makes me more suspicious of interpreting the different station results in Fig. 10.

7) A few minor points: The term 'equivalent barotropic' (p. 6, l. 34) is a dynamical meteorological term and probably not meaningful for ozone (although I understand what you mean). On p. 1, l. 25: 'composition' should be 'deposition'. In Table 1, the last column should be 'average # of soundings'. I thought these were the total number of soundings until I saw Fig. 1.

---

## Author Comment (AC1) · 29 May 2017

We want to thank the reviewer for the very thoughtful and constructive comments.

Comment 1:

The data from Eureka, Ny Aalesund, and Lerwick are homogenized by the institutions of the 3 persons mentioned in the acknowledgements. The other stations are homogenized by the institutions of the authors of the paper. In the revised paper we will include this and some more details about the homogenization procedure. We will also describe which data that are available from the websites.

Unfortunately, changes in ozonesonde types and (vertical) resolutions are not clear-cut. There are often transition periods lasting several years where a combination of

different sondes and resolutions are used. In the revised version we will include the main changes in Table 1 or indicate them in Fig. 1.

Comment 2:

Probably the biggest differences between Bayesian and sequential methods are that in the Bayesian approach the parameters of the model can be seen as random variables and that the Bayesian approach can systematically include prior information (not used in the present study). The Bayesian method is also often seen as more philosophically appealing.

However, the main practical advantage of the Bayesian method (and the reason we use it here) is that we obtain a characteristic ensemble of solutions which systematically provides uncertainties. These uncertainties are not only obtained for all parameters but also for all derived quantities such as trends, annual cycles, differences in annual cycles etc. Another advantage of the model based Bayesian method is that we can use all data directly without first doing, e.g., monthly averaging. As the model is non-linear it is not easy to fit directly, but the Monte-Carlo methods used in Bayesian statistics do this. The fitting could also be done by using a fitting routine that deals with nonlinear models. However, such methods does not allow for directly obtaining the uncertainties.

In the revised version of the paper we will include a somewhat more detailed description of the Bayesian method and the Monte-Carlo procedure used for sampling.

Residuals calculated as the difference between the mean model (cyan in the original Fig. 1) and the original data (black dots in the original Fig. 1) are shown in the attached figures for Ny Aalesund and Lerwick at 500 hPa. In general the residuals are stationary with little low-frequency structure. The distributions are close to symmetric and not far from Gaussian but with some outliers. There is no or only a weak seasonal cycle in the residuals. These results are characteristic for levels below 300 hPa at all stations.

Above 300 hPa an annual cycle is seen in the residuals with largest deviations in the

[Figure]

winter; most probably related to the strong stratospheric variability in this season. In particular at 300 the residuals are positively skewed, probably because this level moves in and out of the stratosphere. In the stratosphere the residuals are again almost Gaussian.

In the revised version we will include the discussion of the residuals (perhaps in a supplement).
* * *
[Figure]

**Ny Aalesund   500**

**Ny Aalesund   500**

**Ny Aalesund   500**

**Fig. 1.** Residuals at 500 hPa. Ny Aalesund. Upper panel: residuals as function of time. Cyan circles are annual means. Middle panel: residuals as function of the day of year. Lower panel: histogram of resid.

[Figure]

**Fig. 2.** As Fig. 1 but for Lerwick.

---

## Author Comment (AC2) · 1 Jun 2017

We want to thank the reviewer for the very thoughtful and constructive comments.

1) We will include a map showing the positions of the stations.

Regarding the data from the Canadian stations, Alert and Resolute, we have only been able to obtain the data from a limited period, 2002-2015. These data are almost always from the winter season. There seem to exist much more data from these stations but we have not been able to retrieve them from the WOUDC or NDACC websites.

We agree that it would strengthen the study to include these stations and we will try again to obtain the data.

[Figure]

2) As mentioned in the response to the other reviewer the main practical advantage of the Bayesian method (and the reason we use it here) is that we obtain a characteristic ensemble of solutions which systematically provides uncertainties. These uncertainties are not only obtained for all parameters but also for all derived quantities such as trends, annual cycles, differences in annual cycles etc. Another advantage of the model based method is that we can use all data directly without first doing, e.g., monthly averaging.

In the revised version of the paper we will include a more detailed description of the Bayesian method and the Monte-Carlo procedure used for sampling.

3) We have calculated the annual cycle of the tropopause height (in pressure) at the four stations with most data. We have used the temperature definition (lapse rate) and calculated the tropopauses from the ozone-sonde records. There is in general a large scatter in the found tropopauses but for the four stations an annual cycle (monthly) can be seen (attached Fig. 1). The general structure - low tropopause in spring and high in autumn - is the same as reported in, e.g., Zangl and Hionka 2001. There is certainly a connection between the annual cycle in ozone at the upper levels, 100 - 300 hPa, and the annual cycle in the tropopause.

At the lowest levels we agree that the annual cycle must be a combination of in situ processes and transport.

In the revised version of the paper the tropopauses will be included in Fig. 8, the calculation described in the text, and the connection to the annual cycle in ozone discussed. The possible mechanisms at the lowest levels will be mentioned.

4) We will change the time-axes of Figs. 4 and 6. There are probably not enough data to make a distinction between the low-frequency variability in Europe and the Arctic. We agree that the Canadian stations would be good to have included.

It is certainly interesting that for the 5 stations (Scoresbysund, Ny Aalesund, So-

dankyla, Eureka, and Orland) the same form of the low-frequency variability is found both near the surface and in the stratosphere. It would seem that such coherent changes were most easily explained by changes in the circulation, which (at least in winter) couples the stratosphere and the troposphere. We will discuss this a little more in the text, although we find that a deeper analysis of the connection between our results and changes in meteorological parameters falls outside the purpose of the present manuscript.

5) Actually, for Ny Aalesund the 1 % significance levels at 500 hPa in Fig. 10 are found for day 40-90 and around day 200. This is also where the annual cycles and the 95 % error bars are (just) separated in Fig. 9. Again, we agree that the Canadian stations would be good to have included.

By the way, it is probably more correct in the captions to say "statistically different .. to the 99 and 95 % levels" than to say "statistically different .. to the 1 and 5 % levels".

6) We seem to disagree somewhat with the reviewer on this point. The two lowest panels in Fig. 11 have in general the same structure. But we will describe the differences and similarities more detailed in the text.

7) We will avoid "equivalent barotropic" and correct the typos.
* * *
[Figure]

**Fig. 1.** The annual cycles in the thermal tropopauses (hPa) for the four stations with most data. Dashed curves are +-2 sigma.

---

## Author Response (AR1)

The Editor
Atmospheric Chemistry and Physics                              June 30, 2017

RE: acp-2017-327

Dear Editor,

Thanks for obtaining two insightful and thorough reviews of our manuscript.

We already submitted replies to most or all of the reviewers' comments in the open discussion. Below we give slightly updated replies and describe in more details how we have changed the manuscript. See also the attached version of the manuscript with track-changes.

Yours Sincerely

Bo Christiansen
Danish Climate Centre
Danish Meteorological Institute
Lyngbyvej 100
DK-2100 Copenhagen O, Denmark
Phone: +45 39 15 74 29 Fax: +45 39 15 74 60
e-mail: boc@dmi.dk

**Review 1:**

We would like to thank the reviewer for a very constructive review.

**Major comments:**

1) We have included the following sentence in section 2.1 (page 5, l14) "The Danish, Norwegian, and Finnish stations were homogenized by authors of the present paper, while the data from Lerwick, Ny Ålesund, and Eureka were homogenized locally (see Acknowledgements)".

Figure 2 now shows the type of ozonesonde used. Red dots indicate EnSci type sondes and black dots Science Pump sondes. For the Canadian stations (Alert and Resolute are now included in the supplementary material) there is no such information in the records before 2000.

Unfortunately, the sensing solution information is not available in the data files for all stations. We have therefore chosen not to include it in the present paper.

The focus of the present paper is on the analysis and not so much on the homogenization. We therefore prefer not to include too much technical discussion on this subject but to leave the discussion of the technical details to a later manuscript. However, we have expanded the relevant paragraph in section 2.1 (page 4) a bit.

2) Probably the biggest differences between Bayesian and sequential method are that in the Bayesian approach the parameters of the model can be seen as random variables and that the Bayesian approach can systematically include prior information (not used in the present study). The Bayesian method is also often seen as more philosophically appealing.

However, the main practical advantage of the Bayesian method (and the reason we use it here) is that we obtain a characteristic ensemble of solutions which systematically provides uncertainties. These uncertainties are not only obtained for all parameters but also for all derived quantities such as trends, annual cycles, differences in annual cycles etc. Another advantage of the model based method is that we can use all data directly without first doing, e.g., monthly averaging.

In the revised version of the paper we have included a more detailed description in section 2.2 (page 5) of the Bayesian method and the Monte-Carlo procedure used for sampling.

Residuals calculated as the difference between the mean model (cyan in Fig. 4) and the original data (black dots in Fig. 4) are now shown in Fig. S1 in the supplementary material for Ny Aalesund at 500 hPa. In general the residuals are stationary with little low-frequency structure. The distribution of the residuals is almost symmetric and not far from Gaussian but with some outliers. There is no or only a weak seasonal cycle in the residuals. These results are characteristic for levels below 300 hPa at all stations.

Above 300 hPa an annual cycle is seen in the residuals with largest deviations in the winter most probably related to the strong stratospheric variability in this season. In particular at 300 the residuals are positively skewed, probably because this level moves in and out of the stratosphere. In the stratosphere the residuals are again almost Gaussian.

The discussion of the residuals is included in the text in section 3.1 on page 6.

**Review 2:**

We would like to thank the reviewer for a very constructive review.

**Major comments:**

1) We have included a new Fig. 1 showing a map of the positions of the stations.

We found the data from the Canadian stations, Alert and Resolute! These are now also analyzed but the figures (S2 and S3) are referred to the supplementary information. Results for these stations look very much like the results from Eureka. This is now describe several places in the text and the regional signal in the trends is mentioned in the conclusions.

2) As mentioned in the response to the other reviewer the main practical advantage of the Bayesian method (and the reason we use it here) is that we obtain a characteristic ensemble of solutions which systematically provides uncertainties. These uncertainties are not only obtained for all parameters but also for all derived quantities such as trends, annual cycles, differences in annual cycles etc. Another advantage of the model based method is that we can use all data directly without first doing, e.g., monthly averaging.

In the revised version of the paper we have included a more detailed description of the Bayesian method and the Monte-Carlo procedure used for sampling (section 2.2, page 5).

3) We have calculated the annual cycle of the tropopause height (in pressure) at the four stations with most data. We have used the temperature definition (lapse rate) and calculated the tropopauses from the information in the ozonesonde records. There is in general a large scatter in the found tropopauses but for the four stations with longest records the tropopauses have been included in Fig. 9 (and for Alert and Resolute in Figs. S2 and S3).

The calculation of the tropopause and its connection to the annual cycle in ozone are now discussed in the text (section 3.2 page 8).

At the lowest levels we agree that the annual cycle must be a combination of in situ processes and transport. This is now briefly mentioned at line 21, page 8.

4) We have changed the time-axes of Figs. 5 and 7.

The three Canadian stations have the same low-frequency variability somewhat different from the European stations. This is now mentioned in the text (page 7, line 28) and in the third bullet point in the Conclusions.

It is certainly interesting that for the 5 stations (Scoresbysund, Ny Aalesund, Sodankyla, Eureka, and Orland) the same form of the low-frequency variability is found both near the surface and in the stratosphere. It would seem that such coherent changes were most easily explained by changes in the circulation, which (at least in winter) couples the stratosphere and the troposphere. We have mentioned this in the text (page 7, last paragraph), although we find that a deeper analysis of the connection between our results and changes in meteorological parameters falls outside the purpose of the present manuscript.

5) Actually, for Ny Ålesund the 1 % significance level at 500 hPa in Fig. 11 is found for day 40-90 and around day 200. This is also where the annual cycles and the 95 % error bars are (just) separated in Fig. 10.

6) We seem to disagree somewhat with the reviewer on this point. The two lowest panels in Fig. 12 have in general the same structure. But we have now described the differences and similarities in a little more details in the text (page 9, line 25).

7) We have deleted "equivalent barotropic" and corrected the typos.

[revised manuscript text omitted]